# Investigating the upper bound of high-frequency electromagnetic waves on unshielded twisted copper pairs

Ergin Dinc [1✉], Syed Sheheryar Bukhari[1], Anas Al Rawi[1,2] & Eloy de Lera Acedo[1]

This paper explores the behaviour of the ubiquitous twisted pairs at high frequencies and wideband excitation of twisted pairs up to 12 GHz. Higher carrier frequencies on twisted pairs can enable the data rates required by the future communication networks; hence, the existing copper infrastructure can be utilised on the last mile complementing the fibre networks. In this paper, we show a fundamental limit on the operating frequency of twisted pairs beyond which twisted pairs start to radiate and behave like an antenna. To validate our theoretical derivations through measurements, we designed a microstrip balun to excite the differential mode on the twisted pairs. At the end, we demonstrate that the standard twisted pairs used in the UK can be used up to 5 GHz carrier frequency without any radiation effect and this upper-bound can be moved to higher frequencies by decreasing the twist lengths.

[1] Department of Physics, Cavendish Laboratory, University of Cambridge, CB3 0HE Cambridge, UK. [2] BT Labs, Adastral Park, Martlesham Heath IP5 3RE, UK.
✉email: ed502@cantab.ac.uk

Delivering high-speed broadband access necessitates replacing old copper infrastructure with fibre optic cables. However, full fibre broadband for everyone is still not a feasible solution due to its high deployment cost especially in the metropolitan and historical cities. In addition, rewiring of densely populated multi-dwelling units and sparsely populated areas are extremely costly and this situation is an important limitation for the fibre deployment. For these reasons, the existing copper infrastructure based on twisted pair (TP) wires will continue complementing the connection between subscribers and the closest fibre-to-the-premises as also discussed by Maes et al.[1]. Thus, it is essential to improve achievable data rates over the copper infrastructure so that the bottlenecks over the last mile can be successfully avoided while satisfying the future data demands.

The advancements in the digital signal processing (DSP) have enabled digital subscriber line (DSL) speeds beyond 100 Mbps as described by Ginis and Cioffi[2]. In addition to the improvements in DSP, the current efforts mostly target increasing the available bandwidth to boost the capacity over TP. The available technology standard, G.fast[3], operates in the frequency spectrum up to 212 MHz and can achieve data rates up to 2 Gbps. The emerging DSL technology, MGFAST[4] aims to increase the available bandwidth by further extending the frequency spectrum up to 848 MHz and targets data rates up to 10 Gbps. As noticed, the available and emerging technologies only consider sub-1 GHz frequency spectrum. We believe that the DSL technologies can further boost the achievable rates beyond 10Gbps by utilising carrier frequencies higher than 1 GHz. In this way, the existing copper infrastructure can be further used to satisfy the data rates required for the future communication networks without any need of replacing all of the copper infrastructure. Towards this objective, this paper investigates electromagnetic (EM) waves on TPs and reports results about the upper-bound on the carrier frequency that can be exploited by the DSL technologies.

TP wires were invented by Alexander Graham Bell[5] in 1881 to reduce radiation from cables, lower crosstalk between pairs and provide robustness against EM interference. Several research papers have been published demonstrating these benefits offered by TPs. Interference from radio stations (at 100–500 KHz) on TP wires was modelled by Stolle[6]. Cross-talk between multiple TP cables in the frequency range below 100 MHz was investigated by Paul and McKnight[7] based on the equivalent-circuit model of TP transmission lines. The references[8–10] include experimental and theoretical results on the magnetic fields around TP and conclude that lower twist pitch, i.e. the length of a twist, results in lower magnetic field around the wire. Therefore, TPs with lower twist pitch length can be placed closer to each other. Yan et al.[11] proposed a finite-difference time-domain (FDTD) algorithm to estimate the coupling of external EM waves (1.5–6 GHz) to TP cables. In this reference, it is observed that the interference is maximum when the wavelength is half-integer or integer multiple of the twist pitch length.

There is also some existing literature on providing Terabit DSL over the copper wires at millimetre wave and THz frequency spectrum. Wang and Mittleman[12] showed experimentally the existence of a TEM-like low dispersion and low attenuation mode on THz frequencies over a single conductor metal wire. Since the conventional transmission lines, where the currents flow in opposite direction on each pair of TP wires, have very high losses at this frequency spectrum, Cioffi et al.[13] have theoretically showed that the waveguides modes between TPs in metal shielded multi-binder wires may be utilised to achieve Tbps up to 100 m and 100 Gbps up to 300 m by distributing the power on different modes with vectoring. However, the utilised modes on multi-binder wire can be only used at high frequencies and the authors[13] investigated 100–300 GHz spectrum. In a more recent paper[14], 1 Tbps data rate was achieved in a two-wire line with metal sheet up to 10 m distance by using a similar vectoring technique to exploit the modal diversity of the waveguide at 200 GHz. Furthermore, Hejazi et al.[15] investigated the millimetre-wave spectrum in TP cables by using transformation optics based numerical simulations to calculate the properties of the waveguide modes and their spatial harmonics. This paper also showed that 1 Tbps can be achieved over a twisted-pair with a plastic sheath up to 10 m. Although there are several papers on investigating millimetre wave and THz frequency spectrum over the copper network, experimental work is still at its infancy. There is still a room for improvement at 1–10 GHz spectrum, where the current DSL technologies have not been used. Therefore, we investigate the fundamental limits in terms of carrier frequency that can be efficiently utilised under 15 GHz.

EM modelling of twisted and untwisted pairs has been extensively studied in the literature. However, parallel with the existing DSL technologies, the papers on modelling EM waves on TP mostly focus on sub-1 GHz frequency spectrum and use relevant approximations that makes them invalid for higher frequencies. In the 1950s, helically wound conductors became popular for travelling-wave tubes to amplify RF signals thanks to their wide bandwidth levels. As a result, there is a large amount of research on modelling helically wound structures, which has a very close geometry with TP. The very first analytical model for a single helical conductor was presented by Sensiper[16] for a finite width tape around a cylindrical geometry. This geometry was specifically selected such that variables can be separated in the cylindrical coordinate system. The exact modelling of the single tape helix model (including dielectric rods near the helix) was proposed by Chernin et al.[17], where the field equations and dispersion relation were derived. Cross-wound double helices, the closest geometry to TP, was investigated by Chodorow and Chu[18]. However, this reference investigated symmetric mode on the geometry where the current directions are the same on both conductors, whereas the existing DSL technologies use asymmetric mode, where the currents are flowing in opposite directions. In this way, the EM field is concentrated between the wires such that this mode is less affected by the objects in the close proximity and enables placement of multiple wires in a confined space. More details on the symmetric and asymmetric modes on untwisted pairs has been recently investigated by Molnar et al.[19]. At the end, the modelling of TPs at higher frequency ranges (>1 GHz) stands as a significant open research problem.

Experimental investigation of higher frequencies on TPs also requires a review of baluns that are utilised for exciting asymmetric mode. The asymmetric mode is also known as differential mode, i.e. the fundamental transverse electromagnetic mode (TEM) on untwisted pair and TP. Most of the state-of-the-art baluns used in the DSL networks are based on classical transformers and generally operate on sub-1 GHz frequencies. This type of baluns provides impedance transformation from 50 Ω input impedance to 100 Ω differential impedance. However, especially at higher frequencies, the classical transformer type baluns do not provide an efficient impedance matching as investigated by Schaich et al.[20]. In this reference, the authors improved the transmission loss up to 10 dB by adding a matching network based on lumped circuit elements. A differential mode launcher based on lumped circuit elements is not an option for us because classical lumped circuit elements have frequency-dependent characteristics, which will make wideband impedance matching impossible. The second option is using a monolithic microwave integrated circuit (MMIC)-type balun[21]. MMIC-type circuit components can efficiently operate up to 300 GHz. These types of baluns are widely available on the market at a higher cost than the classical baluns. MMIC-type

baluns also provide 100 Ω differential output impedance, which is required to be matched to the wire as highlighted in ref. [20].

In this work, we show that TP wires start radiating after a certain frequency and this radiation frequency is a theoretical upper bound that can be used for data communication. To the best of our knowledge, this radiation has not been reported in the literature and the exact radiation frequency depends on wire geometry such as twist pitch length and separation between wires. Therefore, this paper provides in-depth investigation of this effect by providing a theoretical derivation of the radiation frequency as well as numerical and experimental justifications. For our experimental set-up, the transmission curve of the balun should not have any large oscillations across the frequency band of interest. If there are large oscillations, it is likely to miss any radiation effect having similar level of loss with the oscillation magnitude. As will be discussed in the launcher design section, MMIC-type baluns are not suitable for our experimental set-up due to large oscillations on the transmission band. That is why, we also design a microstrip balun that can provide wideband excitation of TPs up to 12 GHz and achieve nearly linear transmission curve. The microstrip balun is referred to as the differential mode launcher in the rest of the paper. Therefore, our theoretical model together with the designed differential mode launcher will enable unlocking high carrier frequencies up to 5–10 GHz, which can provide multiplicative increase in the achievable DSL capacities.

## Results

**Demonstration of a stopband on TP.** We have developed an FDTD simulation in CST Studio Suite[22] in order to validate the proven benefits of TP cables and also observe the propagation of high frequency EM waves. For this purpose, a cable with varying twist pitch length is designed (details of the simulation environment explained in the Supplementary Information—Note 1). The cable starts as an untwisted pair and twisting starts gradually reaching the lowest twist pitch length of 5 mm in the middle section. As noticed in 2D E-field results presented in Fig. 1a for 0.5 and 1.5 GHz, the fields become more confined around the wire when the twist pitch length is lower as also discussed in several references[6,8,9]. More importantly, the TP wire starts radiating after a certain carrier frequency as seen in the E-field results for 7 GHz in Fig. 1a and the $S_{21}$ parameters in Fig. 1b. If TP starts radiating after a certain frequency, it means that there would be an upper-bound on the carrier frequency that can be exploited by the DSL technologies. In the remainder of this section, we provide a theoretical understanding of the radiation from TP and support our claims with numerical simulations and experimental measurements.

**Derivation of EM fields and characteristic equation.** For the derivation of EM fields and the characteristic equation for unshielded TP, we followed the methodology presented in[23]. This is due to the fact that, even though the cable presented in[23] is shielded and geometrically substantially different, the modelling technique in[23] has been verified with numerical simulations and experimental measurements. For this reason, we decided to utilise a similar methodology for the theoretical derivations. Figure 2 presents the TP geometry used in the derivations. The separation between Conductor 1 ($C_1$) and Conductor 2 ($C_2$) is $2r_c$. Twist pitch length, i.e. the full 360° turn, is denoted as $p$ and the twist wavenumber $k_p = 2 \times \pi/p$. Twist angle is found as $\cot(\psi) = k_p r_c$, where $\psi = \pi/2$ is associated with the untwisted pair as $p \to \infty$. In addition, the conductors are assumed as infinitesimal current filaments without a dielectric coating in the theoretical calculations. This assumption helps with the tractability of the derivations while still producing accurate calculations, which are justified with the numerical simulations. Note that, the derivations in this paper investigate the frequency spectrum below 15 GHz; however, the assumptions on the infinitesimal current filaments will not be accurate if the wavelength is comparable with $r_c$. Since the wavelength of 15 GHz (20 mm) is an order of magnitude higher than the investigated $r_c$ values, the derived expressions are able to generate accurate results for this frequency range.

The fields around TP need to satisfy the source-free Maxwell equations

$$\nabla \times \mathbf{E} = i\omega\mu\mathbf{H},  \quad (1)$$

$$\nabla \times \mathbf{H} = (\sigma - i\omega\epsilon)\mathbf{E}.  \quad (2)$$

where the space between the conductors are filled with a medium with conductivity $\sigma$, permittivity $\epsilon$ and permeability $\mu$. $\omega$ is the angular frequency.

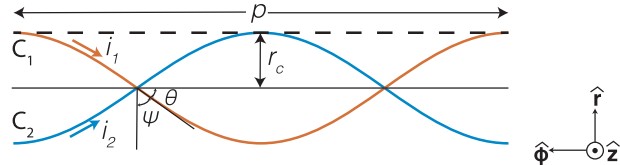

**Fig. 2 TP geometry.** 2D representation of TP wires. Pairs in the wire are denoted as $C_1$ and $C_2$. $p$ is the twist pitch length. $r_c$ is the separation radius between pairs. $\psi$ and $\theta$ depends on $p$ and $r_c$. $i_1$ and $i_2$ are scalar variables representing the current on the pairs. If current variable is positive, it is flowing in the direction showed on the figure.

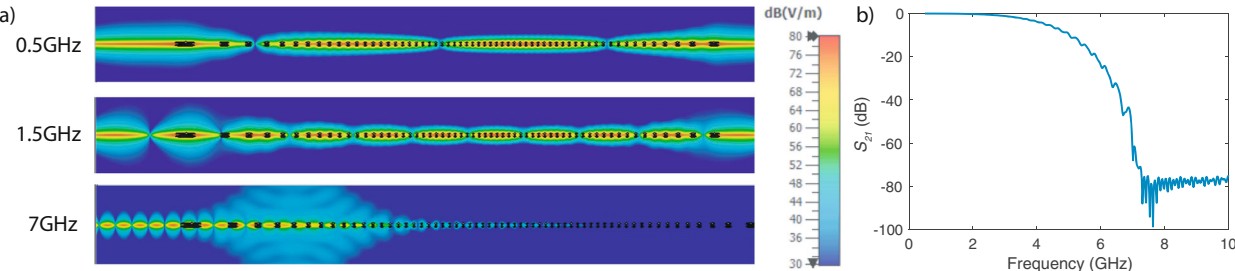

**Fig. 1 Demonstration of radiation effect with a numerical simulation.** Numerical results for a TP with varying twist pitch length (wire length: 90 cm). The investigated wire starts as an untwisted pair and the twist rate is gradually increasing. Lowest twist pitch length of 5 mm is observed close to the middle section. The wire unwinds into an untwisted pair at the end. **a** E-field plots for different frequencies. The fields become more confined for lower twist pitch lengths as in 0.5 and 1.5 GHz. However, the wire radiates at higher frequencies as in the 7 GHz case. **b** $S_{21}$ results for the wire. The effect of radiation can be also seen in the $S_{21}$ results.

The cylindrical coordinate system is used for modelling TP. Therefore, the Helmholtz equation for the axial component of the electric field ($\hat{z}$ axis) can be derived as[23]

$$\nabla^2 E_z + k^2 E_z = 0, \tag{3}$$

where $k = \sqrt{\omega\epsilon\mu}$ is the wavenumber of the surrounding medium. According to the Floquet's theorem, the axial component of the periodic structure can be expressed as[23]

$$E_z(r) = \sum_{m=-\infty}^{\infty} e_{zm}(r)e^{-im(\phi-k_p z)}e^{ih_0 z}, \tag{4}$$

where $(r, \phi, z)$ represent the cylindrical axis system.

The axial electric field component can be further simplified by substituting Eq. (4) into Eq. (3)

$$\frac{1}{r}\frac{d}{dr}\left(r\frac{de_{zm}}{dr}\right) - \left(\gamma_m^2 + \frac{m^2}{r^2}\right)e_{zm} = 0. \tag{5}$$

The lateral wavenumber is $\gamma_m = \sqrt{(h_0 + mk_p)^2 - k^2}$, where $h_0$ is the unknown parameter that will be calculated with the characteristic equation. Eq. (5) obeys the Bessel equation and solutions to the axial field components can be represented with the modified Bessel functions[24]. The axial fields at the centre ($r = 0$) should be finite; thus, the fields for the interior ($r < r_c$) can be expressed with the modified Bessel function of the first kind

$$e_{zm}^{\text{in}}(r) = E_m^{\text{in}}I_m(\gamma_m r), \tag{6}$$

$$h_{zm}^{\text{in}}(r) = H_m^{\text{in}}I_m(\gamma_m r). \tag{7}$$

The fields at the exterior region ($r > r_c$) are vanishing at infinity, so they can be represented as the modified Bessel function of the second kind

$$e_{zm}^{\text{ex}}(r) = E_m^{\text{ex}}K_m(\gamma_m r), \tag{8}$$

$$h_{zm}^{\text{ex}}(r) = H_m^{\text{ex}}K_m(\gamma_m r). \tag{9}$$

The current density on the wires in the Floquet form is written as[23]

$$\mathbf{K}(\phi, z) = \frac{\cos(\psi)\hat{\phi} + \sin(\psi)\hat{z}}{2\pi r_c}\sum_{m=-\infty}^{\infty}(i_1 + (-1)^m i_2)e^{-im(\phi-k_p z)}e^{ih_0 z}, \tag{10}$$

where $i_1$ and $i_2$ are the current flowing through filaments as presented in Fig. 2.

The relationship between the coefficients of the interior and exterior of the fields can be derived by using the following boundary conditions

$$\hat{\mathbf{r}} \times (\mathbf{E}^{ex} - \mathbf{E}^{in}) = 0, \tag{11}$$

$$\hat{\mathbf{r}} \times (\mathbf{H}^{ex} - \mathbf{H}^{in}) = \mathbf{K}, \tag{12}$$

which are valid on the TP filaments at $r = r_c$. In addition, the derivative of the tangential magnetic field is continuous on the boundary.

By applying the boundary conditions, the expressions can be simplified as

$$E_m^{\text{in}} = E_m^{\text{ex}}\frac{K_m(\gamma_m r_c)}{I_m(\gamma_m r_c)}, \tag{13}$$

$$H_m^{\text{in}} = H_m^{\text{ex}}\frac{K_m'(\gamma_m r_c)}{I_m'(\gamma_m r_c)}, \tag{14}$$

where (′) shows the derivative of the modified Bessel functions.

By applying the boundary conditions and using Eqs. (10), (13) and (14), the field coefficients for the exterior region can be derived as

$$E_m^{\text{ex}} = I_m(\gamma_m r_c)\frac{m(h_0 + mk_p)\cos(\psi) + \gamma_m^2 r_c \sin(\psi)}{i2\pi r_c(\sigma - \omega\epsilon)}(i_1 + (-1)^m i_2), \tag{15}$$

$$H_m^{\text{ex}} = \frac{I_m'(\gamma_m r_c)\gamma_m\cos(\psi)}{2\pi}(i_1 + (-1)^m i_2). \tag{16}$$

The other components of the fields can be calculated by using the Hertz vectors ($\mathbf{Z}_E$, $\mathbf{Z}_H$). The Hertz vectors satisfy the Helmholtz equation and can be represented as

$$\mathbf{Z}_E^m = -\frac{\mathbf{E}_z^m}{\gamma_m^2}, \tag{17}$$

$$\mathbf{Z}_H^m = -\frac{\mathbf{H}_z^m}{\gamma_m^2}. \tag{18}$$

At the end, the remaining field components can be found in terms of the Hertz vectors as[25]

$$\mathbf{E}_r = \frac{\partial^2 \mathbf{Z}_E}{\partial r\partial z} - \frac{1}{cr}\frac{\partial^2 \mathbf{Z}_H}{\partial\phi\partial t}, \tag{19}$$

$$\mathbf{E}_\phi = \frac{1}{r}\frac{\partial^2 \mathbf{Z}_E}{\partial\phi\partial z} + \frac{1}{c}\frac{\partial^2 \mathbf{Z}_H}{\partial r\partial t}, \tag{20}$$

$$\mathbf{H}_r = \frac{\partial^2 \mathbf{Z}_M}{\partial r\partial z} + \frac{1}{cr}\frac{\partial^2 \mathbf{Z}_E}{\partial\phi\partial t}, \tag{21}$$

$$\mathbf{H}_\phi = \frac{1}{r}\frac{\partial^2 \mathbf{Z}_M}{\partial\phi\partial z} - \frac{1}{c}\frac{\partial^2 \mathbf{Z}_E}{\partial r\partial t}. \tag{22}$$

The tangential component of the electric field is required to be vanishing on the TP filaments ($r = r_c$); thus, we can write the following

$$(\cos(\psi)\hat{\phi} + \sin(\psi)\hat{z}).\mathbf{E} = 0. \tag{23}$$

By using the field equations for the exterior of the wire at $r = r_c$, the characteristic equation can be derived as

$$\sum_{m=-\infty}^{\infty}\begin{bmatrix} 1 & (-1)^m \\ (-1)^m & 1 \end{bmatrix}\begin{bmatrix} i_1 \\ i_2 \end{bmatrix}S_m(\omega, h_0) = \mathbf{0}, \tag{24}$$

where

$$S_m(\omega, h_0) = (m(h_0 + mk_p) + \gamma_m^2 r_c \tan(\psi))^2 \\ + (kr_c^2\omega\epsilon\gamma_m)\frac{K_m'(\gamma_m r_c)}{I_m'(\gamma_m r_c)}K_m(\gamma_m r_c)I_m(\gamma_m r_c). \tag{25}$$

In the differential mode, the currents on TP flow in opposite directions such that $i_1 = -i_2$. Therefore, the current density can be nonzero for only odd multiples of $m$. The even multiples of $m$ are associated with the symmetrical mode, which is also known as the surface wave mode.

In order to predict the radiation from the cable, we can check the radial component of the Poynting vector. If there is a power flow in the radial direction, it means the wire is behaving like an antenna such that some of the power will be leaking away from TP. For this purpose, the radial component of the Poynting vector is calculated as

$$\langle S_r \rangle = \frac{1}{\mu}Re[(\mathbf{E}\times\mathbf{H}^*).\hat{r}] = \frac{1}{\mu}Re[E_\phi B_z^* - E_z B_\phi^*]. \tag{26}$$

where $Re(x)$ returns the real part of $x$.

By substituting Eqs. (8) and (9) into Eqs. (17), (18), (20) and (22), the radial component of the Poynting vector can be

calculated as

$$\langle (S_r)_m \rangle = \frac{1}{\mu} \mathrm{Re} \left[ \frac{i\omega}{c\gamma_m^2} B_m^{\mathrm{ex}} B_m^{\mathrm{ex}*} K_m'(\gamma_m r) K_m^*(\gamma_m r) \right.$$
$$\left. + \frac{i\omega}{c\gamma_m^2} A_m^{\mathrm{ex}} A_m^{\mathrm{ex}*} K_m(\gamma_m r) K_m'^{*}(\gamma_m r) \right]. \tag{27}$$

As noticed $\langle (S_r)_m \rangle$ will be zero when $(\gamma_m)$ is real-valued as the term inside the $\mathrm{Re}(.)$ function will be purely imaginary. However, if $\gamma_m$ is imaginary, the radial component of the Poynting vector will be nonzero and this causes the TP to radiate. $\gamma_m$ $(\gamma_m = \sqrt{(h_0 + mk_p)^2 - k^2})$ can be imaginary for $m = -1$ beyond a certain frequency range, which is associated with the radiation effect. The lower indexes ($m < -1$) may also contribute to the radiation, but the magnitude of $\langle (S_r)_m \rangle$ reduces with the square of $\gamma_m$. For this reason, the most significant contributor of the radiation is the $m = -1$ index. To show this relationship, we solved Eq. (24) by using a simplifying assumption as in[25] such that the fundamental component $m = 1$ of the differential mode is assumed as the only contributing index. We can solve the

characteristic equation (24) for $m = 1$ numerically in MATLAB (fzero function) by assuming $\gamma_1$ is real. This is a valid assumption as the losses are not included in the theoretical calculations. Then $\gamma_{-1}$ term can be derived as $\gamma_{-1} = \gamma_1 - 2 \times k_p$. Note that, the currents are in the opposite direction ($i_1 = -i_2$) and the differential modes are represented by the odd multiples of $m$; hence, the root of $S_1(\omega, h_0) = 0$ is calculated based on Eq. (25).

The radiation effect can be interpreted with the leaky wave antenna theory[26] as well. Even though the fundamental mode ($m = 1$) is a slow wave, the $m = -1$ space harmonics is a fast wave with $h_{-1} = h_0 - k_p < k$, and this causes the system to behave like an antenna. This situation can be better interpreted from the Brillouin diagram presented in Fig. 3. The Brillouin diagram is plotted for the spectrum between 0.1–15 GHz as our assumptions will not be valid for higher frequency spectrum. $m = -1$ spatial harmonic moves into the radiation zone and the intersection point is associated with $\approx 6$ GHz. As this is a periodic structure with open boundary conditions, the radiation starts as a backfire radiation and the direction of the main lobe of the radiation moves toward to broadside as the frequency increases. The measurement results proving the direction of radiation can be seen in Supplementary Information—Note 5, which also includes additional simulation results demonstrating dominance of the radiation in the observed attenuation. In the investigated spectrum (0.1–15 GHz), $m = -1$ is the only mode contributing to the radiation, and this is a sufficient condition for a leaky wave antenna to radiate[27–29]. Thus, the EM wave guided on TP radiates along the wire.

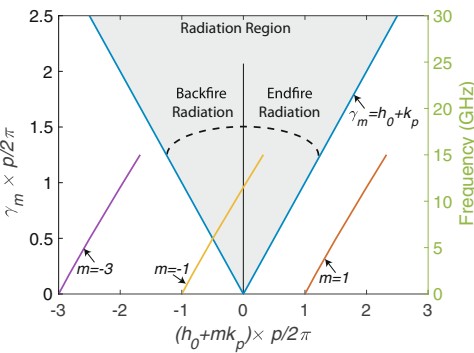

**Fig. 3 Brillouin diagram for the TP.** The plot is for a TP with $p = 25$ mm twist pitch length and shows the $m = 1$, $m = -1$ and $m = -3$ modes. Only the frequency spectrum of 0.1–15 GHz is included in the plot as our assumptions are valid in this frequency range. $m = -1$ moves into the radiation zone after a certain frequency and this is the main reason for observing radiation effect on TP wires. $p$ is the twist pitch length. $k_p = 2 \times \pi/p$. $\gamma_m = \sqrt{(h_0 + mk_p)^2 - k^2}$ is the lateral wavenumber for $m^{th}$ mode, where $k$ is the wavenumber of the surrounding medium and $h_0$ is a parameter calculated from the characteristic equation.

**Analytical and numerical results: upper bound on frequency.** Figure 4a presents the real and imaginary part of $\gamma_{-1}$ for twist pitch length of 15 mm and $r_c = 0.5$ mm. As noticed, the imaginary part of the wavenumber becomes nonzero at frequencies higher than 9.1 GHz for this wire geometry and this frequency is associated with the radiation effect. The radiation frequency can be also estimated by the periodicity of the geometry as $c/(2 \times p)$. This simple estimation predicts 10 GHz for the twist pitch length of 15 mm; however, this estimation is significantly higher than the value calculated with our technique. Therefore, we show that the periodicity of the structure is not enough to estimate the frequency of radiation.

The effects of twist pitch length ($p$) and wire separation radius ($r_c$) are presented in Fig. 4b. The radiation frequency decreases as the twist pitch length increases and the twist pitch length is the

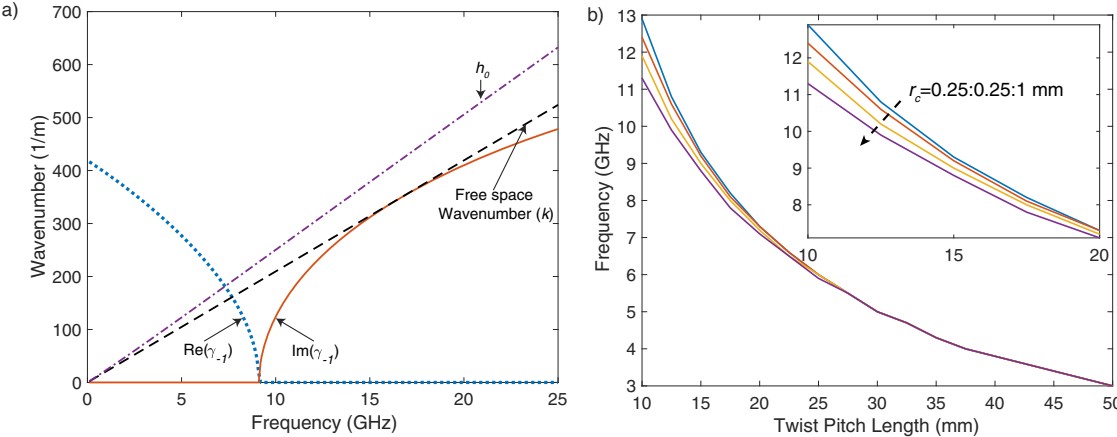

**Fig. 4 Lateral wavenumber and radiation frequency. a** Analytical results for the lateral wavenumber ($\gamma_{-1}$). $h_0$ and $\gamma_{-1}$ are calculated by solving the characteristic equation numerically. The wire starts radiating in the frequency, in which the imaginary part of $\gamma_{-1}$ is non-zero. **b** Radiation frequency results for different wire geometries, where $r_c$ is the separation radius between pairs.

dominant factor, which determines the radiation frequency. Figure 4b also suggests that the radiation frequency is higher for wires that are closely spaced. These radiation frequencies are also upper bounds on the carrier frequency for the differential mode. Communication systems are required to utilise the spectrum below the radiation frequency as the transmitted power beyond this upper bound is radiated to the medium rather than guided along the transmission line.

Table 1 includes the results for different twist pitch lengths and wire separation radius $r_c$. The numerical results are calculated with the FDTD simulations on CST. The radiation frequencies are determined from the $S_{21}$ plots and these frequencies are associated with the start of significant drop on $S_{21}$ levels. As noticed, the developed model is able to predict very close radiation frequency for each geometry. The gap between theory and numerical results are getting larger for lower pitch lengths. The main reason for this is the infinitesimal filament approximation in the theoretical modelling while the wires of TP in the numerical simulations are 3D objects with 0.25 mm radius. Note that, the results presented in this section are for a TP without a dielectric coating, but the dielectric coating only creates a shift in the radiation frequency due to the change in the propagation constants. However, the qualitative behaviour of the cable and radiation effect are similar for both cases as discussed in the

Supplementary Information—Note 3. The details of the simulation environment and excitation of TPs are discussed in the Methods and the next section. In conclusion, the theoretical derivations have shown the fundamental mechanism behind this effect and the radiation frequency can be estimated with the derived characteristic equation.

**Design guidelines for differential mode launcher and simulation environment.** Validating the radiation effect observed in the numerical simulations and our theoretical calculations in the previous section requires a differential mode launcher, which is working up to ≈12 GHz. In this way, we can investigate the radiation effect at twist pitch lengths of 10 mm and higher. The most common telephone wires that are deployed in the UK[30] have twist length of 25 mm complying the BT's Specification of CW1423, but shorter twist lengths of 10–15 mm are commonly used in Category 5–8 cables. That is why, twist lengths of 10–25 mm are investigated in this paper.

We designed a microwave balun as a differential mode launcher as seen in Fig. 5a. The main purpose of this design is to provide a nearly linear transmission response over wideband of frequencies such that the existence of radiation can be proven in an experimental setup. This design may be used in the future DSL networks to extend the frequency spectrum of the existing standards because the proposed launcher design alleviates parasitic losses of the baluns based on the lumped circuit elements and provide flatter response than the existing MMIC-type baluns. The DM launcher starts as a microstrip line matched to 50 Ω. Both top and bottom traces are tapered down to provide a smooth impedance transformation from 50 Ω to the impedance of the connected wire. The narrow end (right side of the DM launcher in Fig. 5a) can be interpreted as double lines with a dielectric layer between them; hence, the supported mode at the end of the DM launcher and double wire are very similar. That is why, it is possible to achieve high launching efficiencies with the proposed microwave balun. Individual wires of TP are separated from each other in order to solder them to the top and bottom traces of the DM launcher as in the bottom sketch of Fig. 5a. Thus, the part of the TP, which is close to the launcher, is modelled as a double line in the launcher design.

**Table 1 Theoretical and numerical results for the radiation frequency.**

| Separation radius ($r_c$) (mm) | Twist length ($p$) (mm) | Radiation frequency (theory) (GHz) | Radiation frequency (simulation) (GHz) |
|---|---|---|---|
| 0.5 | 10 | 12.4 | 13.62 |
| 0.5 | 15 | 9.2 | 9.3 |
| 0.5 | 20 | 7.3 | 7.1 |
| 0.5 | 25 | 5.95 | 5.85 |
| 1 | 10 | 11.2 | 12 |
| 1 | 15 | 8.8 | 8.77 |
| 1 | 20 | 7.1 | 6.7 |
| 1 | 25 | 5.85 | 5.66 |

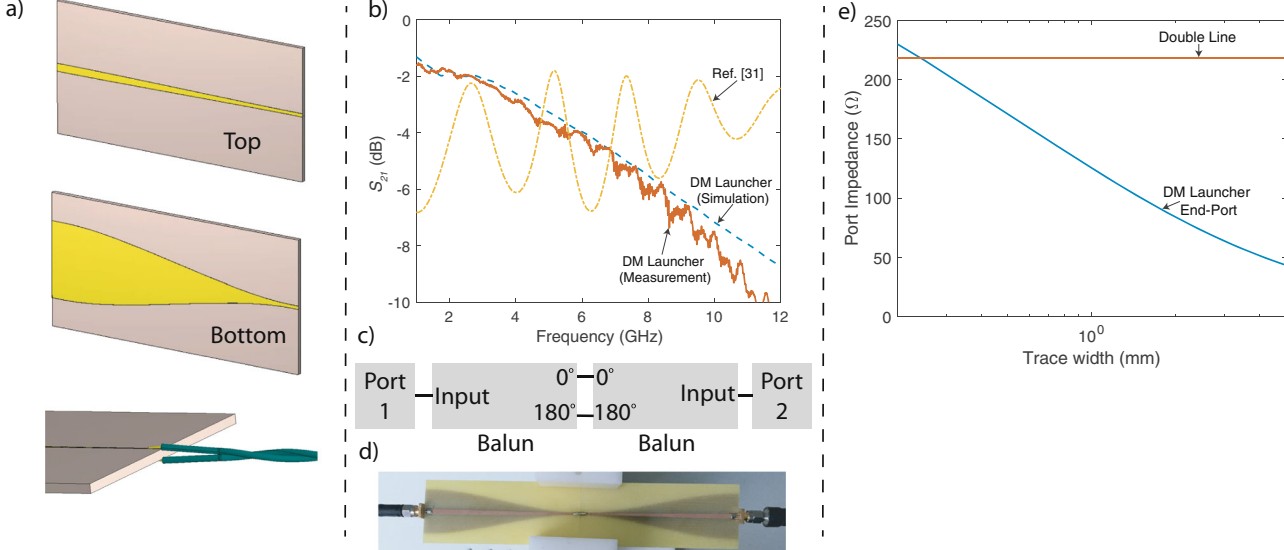

**Fig. 5 DM launcher design. a** Design of DM launcher, **b** transmission loss of back-to-back connected baluns, **c** block diagram the back-to-back connected baluns, **d** measurement set-up, **e** port impedances for the end of DM launcher and double line.

The fundamental mode on the end of the DM launcher and the double wire are both quasi-TEM and the port impedances are nearly constant across the frequency spectrum. Therefore, matching the impedance of these two ports will give the highest performance. Figure 5e includes the port impedance values for the end of the DM launcher and double wire. The port impedances are calculated with the frequency-domain solver of CST[22]. The following parameters are used for the double line: conductor radius 0.25 mm, dielectric thickness 0.25 mm, $\epsilon_r = 2.7$ and centre-to-centre separation 2.18 mm (This value is higher than the TP's centre-to-centre separation distance of 1mm due to additional space for soldering as seen in Fig. 5a). The DM launcher is designed on an FR-4 substrate (thickness 1.6 mm, $\epsilon_r = 4.3$ and $\tan\delta = 0.021$). As noticed from the figure, the best match is achieved when the end trace width of the launcher is 0.2 mm. However, this trace width is not practical as it was not possible to solder wires with conductor radius of 0.25 mm without extending the trace width. For this reason, we fabricate DM launchers with 1.25 mm in our experiments. The fabricated launchers have an impedance mismatch, but they are still good enough to observe the radiation effect on TPs and validate our theory. The width and length of the substrate along with the type of the taper is optimised by using the time-domain solver of CST (The details of these parameters are included in the Supplementary Information—Note 2). Substrate width and length are selected as 50mm and 100mm, respectively. The taper is designed as a raised-cosine taper with ($\beta = 0.08$). The substrate length is especially important for the return loss of lower frequencies as the smoothness of the taper depends on the wavelength. The designed launcher is targeted to operate between 1 and 12 GHz to observe the radiation from the TPs having 10–25 mm twist pitch lengths. Figure 6a includes the photos of the fabricated devices.

In Fig. 5c, d, the block diagram and photo of an end-to-end system without a TP is given. In order to accurately determine the radiation frequency, the back-to-back connected baluns are required to have a nearly linear transmission curve without large oscillations. Otherwise, it is not possible to observe the radiation effect when the radiation loss is on a similar level with the oscillations. This is especially expected for larger twist pitch lengths as will be shown in the next section. For instance, we show a comparison of our launcher with a high-end balun in Fig. 5b. Even though the balun (BALH-0012SSG from Marki Microwave[31]) has a better transmission response at higher frequencies, the sinusoid-like nature of its transmission makes it not practical to use to measure radiation due to aforementioned reasons. Figure 5b also presents the back-to-back $S_{21}$ for the designed DM launcher on FR-4. Note that the transmission loss of this system is equivalent to $-S_{21}$ in dB. Both the numerical

simulation and measurement results suggest that the designed DM launcher on FR-4 has a nearly linear response across 2–12 GHz without any large oscillations. The loss of the proposed DM launcher is higher than the off-the-shelf device, but this is primarily caused by the high dielectric losses introduced by FR-4 substrate, which is preferred due to its easy fabrication, availability and low price. However, an optimised launcher for communication applications can be designed by following the guidelines provided in this section.

**Experimental and numerical results**. The measurement set-up can be seen in Fig. 6b. A custom designed jig is used to stabilise the DM launchers 8–10 cm above the bench such that any proximity effect due to the surrounding objects can be avoided. Twist lengths are not perfectly uniform along the cables and show 1–3 mm variations. The photo of the different twist lengths used in the measurements are included in Fig. 7 and the twist lengths are indicated in the subtitles of each measurement. All measurements are performed for 0.5 m cables. The cables have properties of 0.5 mm diameter copper conductor covered with a 0.25 mm thick cylindrical dielectric. The dielectric properties of the wire are assumed as $\epsilon_r = 2.7$ and $\tan\delta = 0.01$. In order to prove that the received signal is caused solely by the guided waves, we have performed additional measurements by placing one of the launchers in an anechoic laboratory environment and observe very similar measurement results (Supplementary Information—Note 4).

Figure 7 includes the S-parameters for the numerical simulations and the experimental measurements, which are performed with the designed DM launcher. Both $S_{11}$ and $S_{21}$ for different wire pairs are highly consistent with the measurements. $S_{11}$ results show that the EM power is successfully enters the system. This transmitted power cannot reach to the second port beyond the radiation frequency as noticed in the $S_{21}$ results. The same radiation frequency calculated in the numerical simulations are observed in the experimental measurements. There is a small mismatch in the radiation frequency of Wire 2 in Fig. 7b. This is probably caused by the uneven twist length along the wire. In addition, accurate measurement of exact twist lengths becomes harder as the dimensions are small.

The power loss due to radiation effect depends on the twist length as well. Wire 4 ($p = 10$ mm) experiences >30 dB decrease in the transmitted power after 11 GHz. The additional loss in $S_{21}$ is lower for lower twist rates, but even Wire 1 ($p = 25$ mm) has additional 3 dB loss over 0.5 m. Since the TP behaves like a leaky-wave antenna at these frequencies, the transmission loss in dB ($-S_{21}$ dB) linearly scales with the length of the wire. This radiation effect is also present in more practical scenarios such as

a)

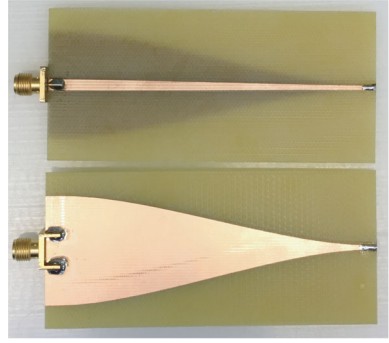

b)

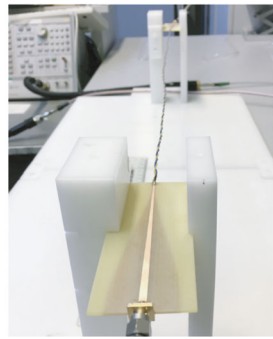

**Fig. 6 Fabricated DM launcher. a** Photos of fabricated DM launcher and **b** measurement set-up.

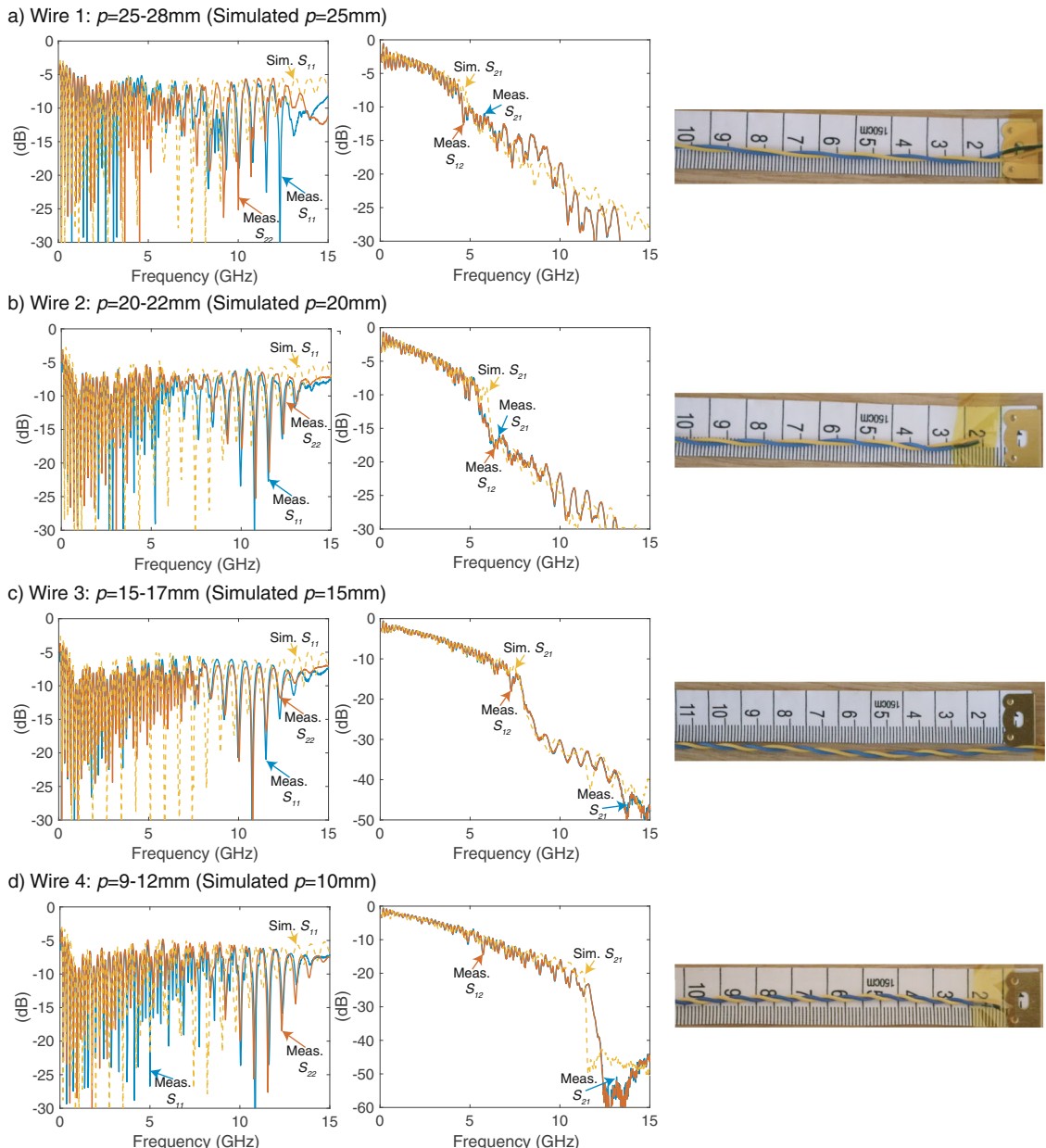

**Fig. 7 Experimental and numerical S-parameters of the end-to-end system.** The results are presented for different twist pitch lengths (*p*) in a decreasing order from subfigure (**a**–**d**). The wire length is 0.5 m in all cases. As noticed, each wire has an increased transmission loss after a certain frequency due to the radiating mode. This frequency moves to higher frequency spectrum for lower twist pitch lengths. Curves and legend keys are in the same colour.

bended wire and multiple TP binders as presented in Supplementary Information—Notes 6 and 7, respectively. Therefore, we can conclude that any future DSL technology, which will operate at high-carrier frequencies, needs to operate below the calculated radiation frequency. Above this frequency, the transmitted power will be radiated away from the line and cause interference in the communication systems nearby.

## Discussion

This paper presents two major contributions towards enabling the exploitation of higher carrier frequencies in the DSL technologies. First, EM fields and characteristic equations of TPs are derived for a helically wound double infinitesimal current filaments. This approach generated an analytically tractable characteristic equation. The individual components of the electric and magnetic fields are calculated. By using these derivations, we

calculated the radial component of the Poynting vector. It is found that $m = -1$ space harmonics of TP has a nonzero radial power flow away from the transmission line after a certain frequency depending on the wire geometry. This means that TP behaves like an antenna above a certain carrier frequency and any communication system that will be designed with these cables needs to operate below this. These results are also validated with numerical simulations and experimental measurements. All of the measurements show excellent consistency with each other. To the best of our knowledge, high frequency modelling of unshielded TPs and radiation effects have not been reported in the literature before. Hence, the findings of this paper will be design guidelines for engineers in designing next generation DSL systems operating at higher carrier frequencies.

The second contribution of this paper is the design of a wideband differential mode launcher that can excite TP in the

frequency spectrum of 1–12 GHz. Unlike off-the-shelf devices, the proposed design has a nearly linear transmission curve across the frequency spectrum, which makes the detection of any radiation loss possible. The differential mode launcher used in our experimental setup is designed for validating our theory. Therefore, it was not perfectly optimised for operations requiring low-losses. The complete design guideline of the differential mode launcher is provided in the paper. In conclusion, we believe that our results and design guidelines will help scientists and engineers to better understand the wave propagation on TPs and enable them to design wideband communication systems operating at higher carrier frequencies.

## Methods

**Launcher prototyping and measurements.** DM launchers were fabricated with 2D milling on FR-4 substrate (substrate thickness 1.6 mm and copper thickness 35 μm on both sides). The TP used in the measurements is mostly known as jumper wire complying the BT's Specification of CW1423[30]. The wire has a single-strand copper conductor of 0.5 mm diameter and dielectric thickness of 0.25 mm. Twist pitch length of the wire is in the range of 25–28 mm (Wire 4 in Fig. 7). The wires with different twist pitch lengths (Wires 1–3) are produced by increasing the number of twists of the cable. We measured the length of all twists along the wire in order to determine the twist pitch length ranges that are stated in Fig. 7. All wires used in the measurements are 0.5 m. S-parameter measurements were performed with a Vector Network Analyser (VNA—8722D from Agilent Technologies). The VNA was calibrated in the range of 50 MHz–20 GHz with the calibration kit of 85052D.

**Numerical simulations.** For the design and characterisation of DM launchers, CST Studio Suite[22] is utilised. CST Studio Suite has both time-domain and frequency-domain solvers exploiting FDTD and Finite Element Method (FEM), respectively. CST's time-domain solver is especially powerful in analysing broadband response of the end-to-end system and we utilise the time domain solver to calculate the numerical S-parameters. In the simulations, TP is excited by using the designed DM launchers. The frequency-domain solver of CST is much faster in calculating the modes on 2D cross section; thus, we mostly used the frequency domain simulations to determine the width of the traces on the DM launcher design. In the simulations, the following material parameters are used: Copper for all conductors ($5.8 \times 10^7$ S/m), FR-4 substrate for DM launcher ($\epsilon_r = 4.3$, $\tan \delta = 0.021$[32], and dielectric around the wire ($\epsilon_r = 2.7$, $\tan \delta = 0.01$).

## Data availability

The data generated in this study have been deposited in the figshare database. The data sets[33] include the numerical simulation and measurement results presented in this paper and its Supplementary Information.

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

## Acknowledgements

This work was supported by the Royal Society Grant IF170002 and Huawei Technologies Dusseldorf GMBH with additional funds provided by BT plc. In addition, E.d.L.A. is funded by Science and Technology Facilities Council (STFC). The authors thank the Royal Society, STFC, Huawei Technologies and BT for these funds. We would also like to show our gratitude to the members of the Fast-Cu Project team for the fruitful discussions and their comments on our work with special thanks to Tobias Schaich for his detailed feedback. The authors also thank John Ely for the fabrication of the DM Launchers.

## Author contributions

E.D. conceived the theory, performed the numerical simulations and designed the differential mode launchers. E.D. and S.S.B. performed the measurements and analysed the data. E.D. wrote the manuscript. All authors contributed to revision of the manuscript. A.A.R. and E.d.L.A. supervised the project.

## Competing interests

The authors declare no competing interests.
