## [Peer Review File · Nature Communications]

Reviewers' Comments:

Reviewer #1:

Remarks to the Author:

The manuscript reveals research on the propagation of high-frequency electromagnetic waves over twisted copper pairs. The two main results are (1) the message that copper pairs can support about 5 GHz of bandwidth, albeit over short distance, and (2) the development of a novel microstrip balun to facilitate experimental validation.

The measurements do support the message that copper twisted pairs support bandwidths up to about 5 GHz. But the analysis does not support the claim that this is caused by radiation. The designed microstrip balun is noteworthy. Further validation is needed to understand its capability to couple higher order modes into the receiver. A detailed review is provided below.

The work is original. The authors did fail to reference related work on Terabit DSL by John Cioffi and the subsequent work on mmW propagation over copper at other research groups. The reference to Cioffi's seminal work (covering 100-300 GHz) and to the (to our knowledge) latest publication on the topic (covering 5-300 GHz) are:

- J.M. Cioffi et al., "Terabit dsls," IEEE Communications Magazine, vol. 56, no. 11, pp. 152–159, 2018.
- A.M. Hejazi et al., "Calculating Millimeter-Wave Modes of Copper Twisted-Pair Cables Using Transformation Optics," IEEE Access, vol. 9, pp. 52079-52088, 2021.

The results show that propagation is hampered above about 5 GHz, depending on the configuration. This is prominently attributed to radiation. However, very little evidence is provided in support of the claimed role of radiation. The prime indirect evidence seems to come from the 2D plots from numerical simulations presented in Figure 1 a). However, based on the presented Figure 1 a), it may well be caused by boundary conditions in the simulation setup. Furthermore, it is unclear whether the pitch length is varied adiabatically. Figure 1 a) seems to suggest the pitch length is varied quite rapidly over just a couple of twists. From the observed attenuation in the S21 curves, one cannot conclude whether radiation is a dominant factor. The presented S21 curves don't support the claim for a strict cut-off frequency. Rather, they reveal a sudden increase in attenuation with frequency, followed by a moderate slope. This may be due to mode conversion rather than radiation, or even the change of shape of a given mode with frequency as has been observed in Hejazi et al.. In case of mode conversion, the microstrip balun or the short piece of untwisted wires leading to the balun, may fail to effectively couple higher order modes into the receiver. The discussion in the section on the Poynting vector starts from the assumption that radiation is dominant and then trims down to focus on the $m=-1$ mode as the dominant source of attenuation, without providing proof that the $m=-1$ is actually dominant in the attenuation.

The theoretical derivation of the fields in the twisted pair rely on assumptions that are valid in the low-frequency regime, but that are not guaranteed to be valid in the frequency regime under consideration. The theoretical model and numerical results are cross validated in Table 1. This is based only on the cut-off frequency. A more qualitative comparison across frequency is not provided. Furthermore, a comparison between measurements and simulations is not provided for the pitch length of 25 mm, whereas this pitch length takes a prominent role in the remainder of the paper. Even larger pitch lengths are known to exist in the U.K. copper access network.

The manuscript jumps from a comparison between theory and simulation in the absence of a dielectric to a comparison between theory and simulation in the presence of a dielectric. A missing step is a one-one comparison between the simulations with and without dielectric. This further reduces confidence that the observed effect is due to radiation. More generally, the numerical conditions are not well explained, reducing reproducibility.

The measurements and simulations correspond well, especially below 5 GHz. However, the selected parameter values for e.g. dielectric constant and loss tangent are not well substantiated. It would be good to clarify if those values are taken from literature, or whether the parameter values were 'fitted' to the

simulations. In the latter case, the theory and simulation would correspond well by design. The measurements and the simulations don't correspond well above 5 GHz, especially in the reflection measurements. For ease of interpretation, the measurements should be presented as a loss per meter. A loss per meter can experimentally be obtained by a cut-back method in which a wire stretch is measured, then its length is reduced by cutting back by e.g. a meter, and then report the delta between the two measurements.

In conclusion, the manuscript supports the message that twisted copper pairs support about 5 GHz of bandwidth, and the proposed microstrip balun progresses the state of art. Further work is required to validate whether radiation is the dominant source of attenuation above about 5 GHz, and to validate the ability of the microstrip balun to couple higher order modes into the receiver.

Reviewer #2:

Remarks to the Author:

I really enjoyed reviewing and reading this article regarding the upper bandwidth for TEM modes on twisted pair transmission lines. It is well written, and summarizes and explain points well. It would be helpful early to state that "pitch" means length of the twist; those terms are used throughout and might be best to use only one if they are the same.

The MMIC balun is also interesting and appropriate for such measurements and a contribution. I am not capable (in reasonable review time) of following the E&M equation development, so trust other reviewers' comments there. However, the results very much match my intuition that the twisted pair will be usable in TEM mode only until the wavelengths start to match the local geometry, particularly the inter-wire spacing. This paper shows that, but also shows the tighter twisting extends this another factor of 3 or 4.

You may also want to say up front that the measurements in Figure 1 are for 1/2 meter of cable (it's inferable from end results, but easy to state up front). It might be nice contribution to investigate more than one twisted-pair together in bundle, which is mentioned but not really addressed. For instance, the 4 tp's of ethernet cable.

You might want to reference or cite as different more recent work by Mittleman, Cioffi, and others on the use of higher-order modes (so not TEM, but closer to what this paper calls the symmetric modes or surface waves). They found and provide measurements of bandwidths in the 100 GHz to 500 GHz band do indeed pass also. This immediate paper for review is of course different, but might want to qualify that on the broad statements that carrier frequency can't exceed about 12 GHz "IN TEM MODE."

Reviewer #3:

Remarks to the Author:

1. In table 1, how is the theoretical Radiation frequency computed?
2. The experimental setup needs to consider more challenging scenarios were the twisted pair exhibits bends & turns.
3. How do you ensure that no energy is radiated from the transmitter to the receiver ? In my view, either the receiver or the transmitter should have been placed inside an anechoic chamber.
4. What is the length of the unshielded twisted-pair cable? It is unclear from the paper and thus the contribution of the results is very limited.
5. Why did you not include any results from an actual installation of twisted pair cable from inside a building or even longer distances.
6. What is the maximum range of transmission that can achieved in more practical scenarios beyond a straight line cable case.
7. The DM launcher looks ad-hoc without any rationale presented to support the particular design and its parameters.
8. The literature survey is very dated with very little state-of-the-art research addressed such as work conducted by Cioffi on Terrabit DSL.

RESPONSE LETTER
Nature Communications

Manuscript ID: NCOMMS-21-08966

**High-Frequency Electromagnetic Waves on Unshielded Twisted Pairs:
Upper Bound on Carrier Frequency**

Dear Reviewers:

We would like to thank you for your effort and time spent in reading and commenting on our manuscript. We appreciate the constructive comments and helpful suggestions from all reviewers. We considered and applied all of the comments. We believe that the revised manuscript has been significantly improved.

The detailed point-by-point responses to the reviewers' comments are given below. The revised parts are presented with coloured text (red). In addition, we have prepared an additional supplemental information to include some of the suggested changes about our numerical and experimental setups.

Reviewer 1:

Comment

The manuscript reveals research on the propagation of high-frequency electromagnetic waves over twisted copper pairs. The two main results are (1) the message that copper pairs can support about 5 GHz of bandwidth, albeit over short distance, and (2) the development of a novel microstrip balun to facilitate experimental validation.

The measurements do support the message that copper twisted pairs support bandwidths up to about 5 GHz. But the analysis does not support the claim that this is caused by radiation. The designed microstrip balun is noteworthy. Further validation is needed to understand its capability to couple higher order modes into the receiver. A detailed review is provided below.

Response

Thank you for your comments and suggestions. Please find our detailed responses in Page 4 of this document.

Comment

The work is original. The authors did fail to reference related work on Terabit DSL by John Cioffi and the subsequent work on mmW propagation over copper at other research groups. The reference to Cioffi's seminal work (covering 100-300 GHz) and to the (to our knowledge) latest publication on the topic (covering 5-300 GHz) are:

- *J.M. Cioffi et al., "Terabit dsls," IEEE Communications Magazine, vol. 56, no. 11, pp. 152–159, 2018.*
- *A.M. Hejazi et al., "Calculating Millimeter-Wave Modes of Copper Twisted-Pair Cables Using Transformation Optics," IEEE Access, vol. 9, pp. 52079-52088, 2021.*

Response

Thank you for these suggestions. These papers focus on much higher frequency spectrum than our work; thus, it is substantially different compared to the scenario investigated in our manuscript. However, we agree that these papers should be discussed in the introduction as related works. We have included the suggested articles as well as other references related to Tbps DSL in the revised manuscript introduction (Page 2 first paragraph - Main Document).

Comment

The results show that propagation is hampered above about 5 GHz, depending on the configuration. This is prominently attributed to radiation. However, very little evidence is provided in support of the claimed role of radiation. The prime indirect evidence seems to come from the 2D plots from numerical simulations presented in Figure 1 a). However, based on the presented Figure 1 a), it may well be caused by boundary conditions in the simulation setup. Furthermore, it is unclear whether the pitch length is varied adiabatically. Figure 1 a) seems to suggest the pitch length is varied quite rapidly over just a couple of twists. From the observed attenuation in the S21 curves, one cannot conclude whether radiation is a dominant factor. The presented S21 curves don't support the claim for a strict cut-off frequency. Rather, they reveal a sudden increase in attenuation with frequency, followed by a moderate slope. This may be due to mode conversion rather than radiation, or even the change of shape of a given mode with frequency as has been observed in Hejazi et al.. In case of mode conversion, the microstrip balun or the short piece of untwisted wires leading to the balun, may fail to effectively couple higher order modes into the receiver. The discussion in the section on the Poynting vector starts from the assumption that radiation is dominant and then trims down to focus on the $m=-1$ mode as the dominant source of attenuation, without providing proof that the $m=-1$ is actually dominant in the attenuation.

Response

Thank you for your remark. We have carefully considered your comments and revised the manuscript accordingly to prove that the observed effect is caused by a radiating mode. As noted by the reviewer as well, the twist length in Figure 1(Main Document) is rapidly changed; hence, this structure does not have a certain cut-off frequency as in Figure 7(Main Document) . We have included this figure in order to demonstrate the radiation only exists after certain frequency while the guided fields are more compact at lower frequencies. The first is the effect that we are explaining in this manuscript. The second effect at the low frequencies is a reported fact about these wires References 8-10 (Main Document).

The designed launcher can only support a TEM mode in the frequency spectrum of interest and the flat section of the launcher is included such that TEM mode in the launcher is converted into the TEM mode on the double wires. Then, the wire starts twisting; thus, there will be infinite number of spatial harmonics guided along the TP. According to the leaky-wave antenna theory, if one of these modes are a fast wave, the periodic structure will start radiating. This effect is discussed in several books and references about periodic structures and related references are added to the manuscript in Page 6 second paragraph. These reference clearly indicate that it is sufficient to show that at least one of the spatial harmonics is a fast wave in order to prove that a periodic structure is radiating. This was previously shown by the propagation constant and we

have also added the Brillouin diagram for twist length of 25mm (see Figure 3 - Main document). The figure makes it obvious that only $m = -1$ is in the radiation zone and the direction of radiation is in the backward direction. To prove that this theory is correct, we have measured the direction of radiation with a directional antenna and presented additional results in the Supplementary Information - Section 5.

Comment

The theoretical derivation of the fields in the twisted pair rely on assumptions that are valid in the low-frequency regime, but that are not guaranteed to be valid in the frequency regime under consideration. The theoretical model and numerical results are cross validated in Table 1. This is based only on the cut-off frequency. A more qualitative comparison across frequency is not provided.

Response

Thank you very much for your remark. This paper aims to show the physical upper-bound on TP carrier frequencies and widely used twist pitch lengths (25mm-10mm) have radiation frequencies below 15GHz; hence, the derivations are using assumptions that are valid for the intended frequency range. The only assumption that might be problematic in the higher frequencies is assuming the TP as infinitesimal current carrying filaments. This assumption will be unrealistic when the dimensions of the wires are comparable to the wavelength. This is not a case in the investigated scenario. We have added the following explanation in Page 3 Line 110-113 (Main Document) to clarify this issue:

“Note that, the derivations in this paper investigates the frequency spectrum below 15GHz; however, the assumptions on the infinitesimal current filaments will not be accurate if the wavelength is comparable with r_c . Since the wavelength of 15GHz (20mm) is much higher than used r_c values (0.5mm-1mm) in this paper, the derivations can be used for this frequency range.”

Comment

Furthermore, a comparison between measurements and simulations is not provided for the pitch length of 25 mm, whereas this pitch length takes a prominent role in the remainder of the paper. Even larger pitch lengths are known to exist in the U.K. copper access network.

Response

We guess that the reviewer means the Table 1 comparison between simulation and theory not the comparison between measurements and simulations because 25mm twist pitch length is already included in our results in Figure 7 (Main Document) . To address this comment, we added 25mm

twist length to the Table 1(Main Document) . Higher twist lengths are already included in Fig 4(b)(Main Document) .

Comment

The manuscript jumps from a comparison between theory and simulation in the absence of a dielectric to a comparison between theory and simulation in the presence of a dielectric. A missing step is a one-one comparison between the simulations with and without dielectric. This further reduces confidence that the observed effect is due to radiation.

Response

If the dielectric is removed around the wire, this will slightly change the propagation constants of the modes, but the general behaviour of the wire will stay the same. In order to demonstrate this claim, we added a section into the supplementary information and discuss the effect of dielectric layer by presenting simulation results with and without dielectric layer. Please find this section in “Comparison of Simulation With and Without Dielectric” section in the supplementary information.

Comment

More generally, the numerical conditions are not well explained, reducing reproducibility.

Response

We agree with the reviewer. The previous version included a brief explanation of the simulation settings. Therefore, we added a detailed section about the numerical setup to the supplementary information. Please see “Numerical Simulation Environment” in the supplementary informations.

Comment

The measurements and simulations correspond well, especially below 5 GHz. However, the selected parameter values for e.g. dielectric constant and loss tangent are not well substantiated. It would be good to clarify if those values are taken from literature, or whether the parameter values were ‘fitted’ to the simulations. In the latter case, the theory and simulation would correspond well by design.

Response

It is a very good point. Dielectric properties are very hard to find as various papers report significantly different values along with the variance introduced during the manufacturing process. In our simulations, we chose an acceptable range for the dielectric properties of FR-4 and PVC. Then, the final values are chosen for the best fit to the simulations. However, we kept the initial range

selection narrow enough, so the accuracy of the simulations can be guaranteed. The simulation results for the back-to-back launchers, 0.5m line and 1m line are presented in the paper and the supplementary information. All of our results have very high consistency. We have included the description of our methodology in the second paragraph of the supplementary information “Numerical Simulation Environment” section.

Comment

The measurements and the simulations don't correspond well above 5 GHz, especially in the reflection measurements.

Response

We agree with the reviewer. In the original manuscript, we presented results that were simulated with waveguide ports in CST, but waveguide ports become inaccurate when there is a radiating field in the simulation domain. This is the case for our simulation and S_{11} curves become inconsistent with the measurements when the wire starts radiating. We added in-detail description of the simulation domain in the supplementary information (including the reasoning for simulating with the discrete port). Hence, all of the simulation settings are included in order to increase reproducibility of our results. Fig 7(Main Document) includes the simulation results with discrete ports and all of the curves show very high consistency.

Comment

For ease of interpretation, the measurements should be presented as a loss per meter. A loss per meter can experimentally be obtained by a cut-back method in which a wire stretch is measured, then its length is reduced by cutting back by e.g. a meter, and then report the delta between the two measurements.

Response

We have now included loss/m results calculated with the numerical simulations. Since our numerical simulations and measurements are in good agreement, we preferred generating this result with the numerical simulations. We believe that this result will make it easier for a reader to compare our results with other papers. Thank you very much for this constructive feedback. The related section can be found in the supplementary information.

Comment

In conclusion, the manuscript supports the message that twisted copper pairs support about 5 GHz of bandwidth, and the proposed microstrip balun progresses the state of art. Further work is required to

validate whether radiation is the dominant source of attenuation above about 5 GHz, and to validate the ability of the microstrip balun to couple higher order modes into the receiver.

Response

Thank you for your comments and suggestions. We have tried to implement and include all of them in the revised manuscript and the supplementary information.

Reviewer 2:

Comment

I really enjoyed reviewing and reading this article regarding the upper bandwidth for TEM modes on twisted pair transmission lines. It is well written, and summarizes and explain points well. It would be helpful early to state that "pitch" means length of the twist; those terms are used throughout and might be best to use only one if they are the same.

Response

Thank you for your comment. Yes, they are the same. We realised that there is an inconsistent terminology in the paper about "pitch length" and "twist pitch length". Since the common notation for this is twist pitch length, we revised this term as "twist pitch length" throughout the manuscript to avoid confusion.

Comment

The MMIC balun is also interesting and appropriate for such measurements and a contribution. I am not capable (in reasonable review time) of following the E&M equation development, so trust other reviewers' comments there. However, the results very much match my intuition that the twisted pair will be usable in TEM mode only until the wavelengths start to match the local geometry, particularly the inter-wire spacing. This paper shows that, but also shows the tighter twisting extends this another factor of 3 or 4.

Response

Thank you very much for your positive comments.

Comment

You may also want to say up front that the measurements in Figure 1 are for 1/2 meter of cable (it's inferable from end results, but easy to state up front).

Response

We now make sure that all of the wire lengths are included in the manuscript and the mental files. These wire lengths are also added to the figure captions to make them clear to the reader.

Comment

It might be nice contribution to investigate more than one twisted-pair together in bundle, which is

mentioned but not really addressed. For instance, the 4 tp's of ethernet cable.

Response

We agree with the reviewer. We included results for two TPs and show that this radiation effect exists for each wire in the associated frequencies depending on the wire geometries. These results can be generalised for cables with more pairs.

Comment

You might want to reference or cite as different more recent work by Mittleman, Cioffi, and others on the use of higher-order modes (so not TEM, but closer to what this paper calls the symmetric modes or surface waves). They found and provide measurements of bandwidths in the 100 GHz to 500 GHz band do indeed pass also. This immediate paper for review is of course different, but might want to qualify that on the broad statements that carrier frequency can't exceed about 12 GHz "IN TEM MODE."

Response

Thank you for these suggestions. We agree that these papers should be discussed in the introduction as related works even though they are substantially different compared to our presented work due to the frequency spectrum of interest. We have included several references covering the recent works on Tbps DSL in the revised manuscript introduction (Page 2 first paragraph (Main Document)).

TEM mode can be used after 12GHz, but the losses become extremely high. That's why, we do not expect there would be any practical system that can use TEM mode after 10GHz.

Reviewer 3:

Comment

1. In table 1, how is the theoretical Radiation frequency computed?

Response

Thank you very much for your comments and suggestions. We have revised the explanation in Page 5 Line 162-165 (Main Document). By following this description, this equation can be solved with the `fzero` - Root of nonlinear function of MATLAB or a custom code based on Equation 25. We also provide the data in all of the presented figures for the reproducibility of the results.

“We can solve the characteristic equation (24) for $m = 1$ numerically in MATLAB (`fzero` function) by assuming γ_1 is real. This is a valid assumption as the losses are not included in the theoretical calculations. Then γ_{-1} term can be derived as $\gamma_{-1} = \gamma_1 - 2 \times k_p$. Note that, the currents are in the opposite direction ($i_1 = -i_2$) and the differential modes are represented by the odd multiples of m ; hence, the root of $S_1(\omega, h_0) = 0$ is calculated based on (25). ”

Comment

2. The experimental setup needs to consider more challenging scenarios were the twisted pair exhibits bends & turns.

Response

In this paper, the main objective is to reveal the physical upper-bound on the carrier frequency. Therefore, investigation of practical scenarios is not in the scope of this manuscript. However, we still think that proving the proposed upper-bound exists in practical scenarios is also valuable for the reader. Therefore, we add additional results in the supplementary informations covering the existence of radiation when the cable is bended. The results were very close to the straight wire as differential mode is robust against the bends in the cable.

Comment

3. How do you ensure that no energy is radiated from the transmitter to the receiver ? In my view, either the receiver or the transmitter should have been placed inside an anechoic chamber.

Response

Thank you for your comment. We have added additional results on the supplementary information. The new results include a lab anechoic environment and no transmission line scenarios. In addition,

we investigated multiple twisted pair scenarios, which is very common in the ethernet cables.

Comment

4. *What is the length of the unshielded twisted-pair cable? It is unclear from the paper and thus the contribution of the results is very limited.*

Response

We have made sure that all of the wire lengths are included in the manuscript and the supplementary informations. These wire lengths are also added to the figure captions to make them clear to the reader.

Comment

5. *Why did you not include any results from an actual installation of twisted pair cable from inside a building or even longer distances.*

6. *What is the maximum range of transmission that can achieved in more practical scenarios beyond a straight line cable case.*

Response

Thank you for your remark. In this manuscript, the main objective is to reveal the upper-bound for the TEM mode on twisted pair wires. This has not been reported in the literature before and this manuscript provides theoretical, numerical and experimental justification of this radiation effect. Since the wire starts radiating after a certain frequency, practical deployments cannot go over this spectrum due to the extreme losses. In addition to this, we have provided additional results on the presence of the radiation effect in more practical scenarios (multiple TPs and bended wire). However, tests in an actual installation or calculating the range for a specific application are not in the scope of our current manuscript. This manuscript provides a very fundamental contribution by presenting the radiation effect after a certain frequency range together with its physical origin. Thus, we believe that several research papers investigating the practical scenarios will follow our work.

Comment

7. *The DM launcher looks ad-hoc without any rationale presented to support the particular design and its parameters.*

Response

The DM launcher is necessary to perform our experiments as we cannot use an off-the-shelf

equipment as discussed in the "Design Guidelines for Differential Mode Launcher and Simulation Environment" section. We have added more explanation to this section and the design parameters are further discussed in the supplementary information - Section 2.

Comment

8. The literature survey is very dated with very little state-of-the-art research addressed such as work conducted by Cioffi on Terrabit DSL.

Response

Thank you for these suggestions. We agree that these papers should be discussed in the introduction as related works even though they are substantially different compared to our presented work due to the frequency spectrum of interest. We have included several references covering the recent works on Tbps DSL in the revised manuscript introduction (Page 2 first paragraph).

REVIEWER COMMENTS

Reviewer #1 (Remarks to the Author):

The manuscript reveals research on the propagation of high-frequency electromagnetic waves over twisted copper pairs. The two main results are (1) the message that copper pairs can support about 5 GHz of bandwidth, albeit over short distance, and (2) the development of a novel microstrip balun to facilitate experimental validation.

The measurements do support the message that copper twisted pairs support bandwidths up to about 5 GHz. But the analysis does not support the claim that this is caused by radiation. The designed microstrip balun is noteworthy. Further validation is needed to understand its capability to couple higher order modes into the receiver. A detailed review is provided below.

The work is original. The authors did fail to reference related work on Terabit DSL by John Cioffi and the subsequent work on mmW propagation over copper at other research groups. The reference to Cioffi's seminal work (covering 100-300 GHz) and to the (to our knowledge) latest publication on the topic (covering 5-300 GHz) are:

- J.M. Cioffi et al., "Terabit dsls," IEEE Communications Magazine, vol. 56, no. 11, pp. 152–159, 2018.
- A.M. Hejazi et al., "Calculating Millimeter-Wave Modes of Copper Twisted-Pair Cables Using Transformation Optics," IEEE Access, vol. 9, pp. 52079-52088, 2021.

The results show that propagation is hampered above about 5 GHz, depending on the configuration. This is prominently attributed to radiation. However, very little evidence is provided in support of the claimed role of radiation. The prime indirect evidence seems to come from the 2D plots from numerical simulations presented in Figure 1 a). However, based on the presented Figure 1 a), it may well be caused by boundary conditions in the simulation setup. Furthermore, it is unclear whether the pitch length is varied adiabatically. Figure 1 a) seems to suggest the pitch length is varied quite rapidly over just a couple of twists. From the observed attenuation in the S21 curves, one cannot conclude whether radiation is a dominant factor. The presented S21 curves don't support the claim for a strict cut-off frequency. Rather, they reveal a sudden increase in attenuation with frequency, followed by a moderate slope. This may be due to mode conversion rather than radiation, or even the change of shape of a given mode with frequency as has been observed in Hejazi et al.. In case of mode conversion, the microstrip balun or the short piece of untwisted wires leading to the balun, may fail to effectively couple higher order modes into the receiver. The discussion in the section on the Poynting vector starts from the assumption that radiation is dominant and then trims down to focus on the $m=-1$ mode as the dominant source of attenuation, without providing proof that the $m=-1$ is actually dominant in the attenuation.

The theoretical derivation of the fields in the twisted pair rely on assumptions that are valid in the low-frequency regime, but that are not guaranteed to be valid in the frequency regime under consideration. The theoretical model and numerical results are cross validated in Table 1. This is based only on the cut-off frequency. A more qualitative comparison across frequency is not provided. Furthermore, a comparison between measurements and simulations is not provided for the pitch length of 25 mm, whereas this pitch length takes a prominent role in the remainder of the paper. Even larger pitch lengths are known to exist in the U.K. copper access network.

The manuscript jumps from a comparison between theory and simulation in the absence of a dielectric to a comparison between theory and simulation in the presence of a dielectric. A missing step is a one-one comparison between the simulations with and without dielectric. This further reduces confidence that the observed effect is due to radiation. More generally, the numerical conditions are not well explained, reducing reproducibility.

The measurements and simulations correspond well, especially below 5 GHz. However, the selected parameter values for e.g. dielectric constant and loss tangent are not well substantiated. It would be good to clarify if those values are taken from literature, or whether the parameter values were 'fitted' to the

simulations. In the latter case, the theory and simulation would correspond well by design. The measurements and the simulations don't correspond well above 5 GHz, especially in the reflection measurements. For ease of interpretation, the measurements should be presented as a loss per meter. A loss per meter can experimentally be obtained by a cut-back method in which a wire stretch is measured, then its length is reduced by cutting back by e.g. a meter, and then report the delta between the two measurements.

In conclusion, the manuscript supports the message that twisted copper pairs support about 5 GHz of bandwidth, and the proposed microstrip balun progresses the state of art. Further work is required to validate whether radiation is the dominant source of attenuation above about 5 GHz, and to validate the ability of the microstrip balun to couple higher order modes into the receiver.

Reviewer #2 (Remarks to the Author):

I really enjoyed reviewing and reading this article regarding the upper bandwidth for TEM modes on twisted pair transmission lines. It is well written, and summarizes and explain points well. It would be helpful early to state that "pitch" means length of the twist; those terms are used throughout and might be best to use only one if they are the same.

The MMIC balun is also interesting and appropriate for such measurements and a contribution. I am not capable (in reasonable review time) of following the E&M equation development, so trust other reviewers' comments there. However, the results very much match my intuition that the twisted pair will be usable in TEM mode only until the wavelengths start to match the local geometry, particularly the inter-wire spacing. This paper shows that, but also shows the tighter twisting extends this another factor of 3 or 4.

You may also want to say up front that the measurements in Figure 1 are for 1/2 meter of cable (it's inferable from end results, but easy to state up front). It might be nice contribution to investigate more than one twisted-pair together in bundle, which is mentioned but not really addressed. For instance, the 4 tp's of ethernet cable.

You might want to reference or cite as different more recent work by Mittleman, Cioffi, and others on the use of higher-order modes (so not TEM, but closer to what this paper calls the symmetric modes or surface waves). They found and provide measurements of bandwidths in the 100 GHz to 500 GHz band do indeed pass also. This immediate paper for review is of course different, but might want to qualify that on the broad statements that carrier frequency can't exceed about 12 GHz "IN TEM MODE."

Reviewer #3 (Remarks to the Author):

1. In table 1, how is the theoretical Radiation frequency computed?
2. The experimental setup needs to consider more challenging scenarios were the twisted pair exhibits bends & turns.
3. How do you ensure that no energy is radiated from the transmitter to the receiver ? In my view, either the receiver or the transmitter should have been placed inside an anechoic chamber.
4. What is the length of the unshielded twisted-pair cable? It is unclear from the paper and thus the contribution of the results is very limited.
5. Why did you not include any results from an actual installation of twisted pair cable from inside a building or even longer distances.
6. What is the maximum range of transmission that can achieved in more practical scenarios beyond a straight line cable case.
7. The DM launcher looks ad-hoc without any rationale presented to support the particular design and its parameters.
8. The literature survey is very dated with very little state-of-the-art research addressed such as work

conducted by Cioffi on Terrabit DSL.

RESPONSE LETTER
Nature Communications

Manuscript ID: NCOMMS-21-08966A

**High-Frequency Electromagnetic Waves on Unshielded Twisted Pairs:
Upper Bound on Carrier Frequency**

Dear Reviewers:

We would like to thank you for your effort and time spent in reading and commenting on our revised manuscript. We appreciate the constructive comments and helpful suggestions from all reviewers. We have considered and applied all of them in the revised manuscript. The detailed point-by-point responses to the reviewers' comments are given below. The revised parts are presented with coloured text (red).

Reviewer 1:

Comment

In conclusion, the revised manuscript has improved, and the majority of concerns have been fully or partly addressed as detailed below. Some concerns remain, as also detailed below. Due to the aggressive peer-review timeline, the supplemental information document was not reviewed in great detail.

Response

Thank you very much for your positive acknowledgements on our work. We have tried to apply all of your suggestions in the revised manuscript. Please see our detailed responses below.

Comment

The work is now better positioned versus related work in the adjacent millimeter wave regime.

The authors have detailed the simulation settings, including the dimensions of the bounding box and the range of values for permittivity and loss tangent. This also improves reproducibility of the results. The authors agree in the rebuttal letter that in the simulation setup around Figure 1, the pitch length is not varied adiabatically.

Response

Thank you very much for your remarks.

Comment

The authors provided additional experimental proof of the existence of significant radiation from the wires. While the authors provide evidence of significant radiation, it is claimed but not proven that radiation is dominant in the observed attenuation. This makes that the following concern remains somewhat unresolved “the discussion in the section on the Poynting vector starts from the assumption that radiation is dominant and then trims down to focus on the $m=-1$ mode as the dominant source of attenuation, without providing proof that the $m=-1$ is actually dominant in the attenuation”. Another source of attenuation to be considered is mode conversion. The authors confirm (only) in the rebuttal letter that the mode coupler design is optimized only for the TEM mode, and hence is likely to attenuate higher order modes. It is recommended to state this more explicitly in the main document or in the supplemental information document.

Response

Thank you very much for your comments. We have responded these comments in two parts and

revised the manuscript accordingly.

b) First Part of the comment about the radiation effect being the dominant factor: We agree with the reviewer’s comment. In the revision, we heavily focused on proving that there is a significant amount of radiation, but it is also important to show that this effect is the dominant factor. Thank you very much for pointing this out. We performed new simulations and revised the Section 5 of the Supplementary Information (Figure 7 of the Supplementary Information includes the new simulation results.). If there is an important amount of loss due to mode conversion or resonating modes in the simulated structure, this power will create more losses in the metals and dielectrics. However, the simulation results do not show any sudden increase in the metal and dielectric losses that supports this theory. Please see the added text below or Section 5 of the Supplementary Information.

[Added Paragraph Section 5 of the Supplementary Information:] After proving the presence of the radiation, it is also critical to demonstrate that the radiation is the dominant source of loss for the observed reduction in the S_{21} levels. Other possible explanations for this reduction in the received signal levels might be resonating modes and conversion of the propagating power to these modes. In order to justify our claims, we designed a simulation setup with the designed DM launcher and a TP wire with 15mm twist pitch length and 0.5m length. All material parameters are utilised as in the previous simulations. Figure 7(a) shows the S-parameters for this system and there is a significant reduction in the S_{21} levels after 7.8GHz (This simulation result was also presented in Figure 7 of the main document and this result is included in the supplementary information for comparison with the following results). To calculate losses due to radiation, metals and dielectrics, we placed several monitors to the simulation domain. Radiation monitors are placed from 5GHz to 10GHz with 0.5GHz intervals. Three different values for the distance between the simulation bounding box and the closest simulated structure (L_{Box}) are utilised. Different values are simulated to show that there is no impact from the interaction between the propagating fields in the wire and the open boundary conditions. Figure 7(b) shows the amount of power absorbed by the open boundary conditions (equivalent of perfectly-matched-layer in other simulation softwares). As noticed, there is a more than 4-fold increase in the radiated power at 8GHz, which is consistent with the S-parameters. Furthermore, all three simulated bounding box lengths are converged to almost identical values and this shows the accuracy of the results. Figure 7(c) shows the distribution of the simulated power. Let’s assume that the port 1 is excited in the simulation with an input power of 0.5W. Power outgoing from ports includes both transmitted power to port 2 and reflected power to port 1 during the simulation time. As seen in the figure, the power outgoing from the ports is decreasing with the frequency. This is an expected result as dielectric/metal losses are increasing with frequency and there

is radiating power in the system as proved by previously discussed experiments and simulations. Note that FR-4 is used for the launchers thanks to its ease of fabrication, cost and availability, but it is a very lossy dielectric material². The rapid fluctuations in the power is caused by the high S_{11} values of the DM launcher, which is also visible in the presented S-parameters in Figure 7(a). As noticed in Figure 7(c), the dielectric losses are increasing until 7.5GHz and remain constant after this frequency. Metal losses slightly decrease at 8GHz and remain nearly constant after that. Both of these trends are caused by the radiation as after 7.8GHz there is less amount of power on the wire interacting with metal and dielectric materials. These losses stay constant after the radiation frequency as the signal needs to pass through one of the DM launchers and excite the radiating mode on the TP. If there is an important amount of loss due to mode conversion and resonating modes, we need to observe an increase in dielectric and metal losses as well. In the end, the simulation results prove that the dominant source for the observed effect is the radiation due to the periodicity of the wire as predicted with our theory.

Note: references and figure numbers are from the Supplementary Information.

a) Second Part of the comment about the higher order modes in the DM launcher: If higher frequency ranges are utilised with the current design of the DM launcher, the launcher is likely to support higher order modes and additional losses will be introduced due to the existence of these higher order modes. This is not likely in the frequency range of interest in our work. We have added the below reference to the manuscript, which includes a formula for the cut-off frequency of the first higher order mode in a microstrip line. As a rule of thumb, the thickness of substrate needs to be lower than 1/10 of the wavelength for the microstrip line to support only quasi TEM mode. According to this calculation, the DM launcher may support higher order modes after 31GHz and the operation range of the DM launcher is much lower than this frequency in our manuscript. We added additional explanations to the Supplementary Information in order to clarify this. You may find the added paragraph below or refer to the Section 2 of the Supplementary Information - Red text. In addition, we added some guidelines to avoid higher order modes if one aims to design a similar device for higher frequency ranges.

[Added Paragraph Section 2 of the Supplementary Information:] Note that, the DM launcher is designed by assuming that it only supports the dominant quasi-TEM mode. This is a valid assumption for the frequency range of interest in our work (<12GHz). If this design is utilised for higher frequency ranges without any modification, the DM launcher is likely to support higher order modes and the launching efficiency will be significantly lower due to mode conversion within the launcher. The cut-off frequency of the first higher order mode in microstrip lines are given in [5]. According to this reference, as a rule of thumb, the operation frequency of microstrip line should be much lower than a frequency whose

wavelength is ten times of the thickness of the substrate. For our design, the first higher order mode is expected to appear at around 31GHz which is much higher than our frequency range of interest. Hence, a designer should be careful about avoiding higher order modes while designing a DM launcher for different frequency range of operation. The operating frequency of DM launcher can be increased by simply using a substrate with lower thickness.

[5 in the Supplementary Information] Bahl, I. J., Trivedi, D. K. A Designer's Guide to Microstrip Line (Microwaves, 1977), 174–182.

Comment

In the supplemental information, the authors have provided additional graphs that allow the reader to evaluate correspondence between measurements and simulations. The authors have also provided the missing step of a direct comparison between simulations with and without dielectric.

Response

Thank you very much for your comment.

Comment

Loss per meter results are provided using a cut-back method. However, the results are only based on simulation, not on measurements.

Response

Thank you for your remark. Since the measurement and simulation results show very high consistency for different wire settings, we preferred to give these results only based on simulations.

Reviewer 2:

Comment

Nice work - good luck with all.

Response

Thank you very much for your time spent in reviewing our manuscript. We really appreciate all of your suggestions that helped us to improve our manuscript.

Reviewer 3:

Comment

I have enjoyed reading the revised manuscript and

Response

Thank you very much for your efforts in reading and commenting on our manuscript. We really appreciate all of your comments that enable us to improve our manuscript.

REVIEWERS' COMMENTS

Reviewer #1 (Remarks to the Author):

The authors have added sufficient context to the manuscript and supplemental document to aid in interpreting the results.

RESPONSE LETTER
Nature Communications

Manuscript ID: NCOMMS-21-08966B

**High-Frequency Electromagnetic Waves on Unshielded Twisted Pairs and
Upper Bound on Carrier Frequency**

Reviewer 1:

Comment

The authors have added sufficient context to the manuscript and supplemental document to aid in interpreting the results.

Response

Thank you very much for your positive acknowledgements on our work. We really appreciate your time spent on reviewing our work.